

# Late Cenozoic Sea Surface Temperature evolution of the South Atlantic Ocean

Frida S. Hoem[1], Adrián López-Quirós[2, 3], Suzanna van de Lagemaat[1], Johan Etourneau[4,5], Marie-Alexandrine Sicre[4, 6], Carlota Escutia[7], Henk Brinkhuis[1,8], Francien Peterse[1], Francesca Sangiorgi[1], Peter K. Bijl[1]

[1]Department of Earth Science, Utrecht University, Utrecht, the Netherlands
[2]Department of Stratigraphy and Paleontology, University of Granada, Granada, Spain
[3]Department of Geoscience/iCLIMATE Centre, Aarhus University, Aarhus C, Denmark
[4]EPHE, PSL Research University, Paris, France.
[5]University of Bordeaux, CNRS, Pessac, France
[6]LOCEAN, CNRS, Sorbonne Université, Campus Pierre et Marie Curie, Paris, France
[7]IACT, CSIC-Universidad de Granada, Granada, Spain
[8]Department of Ocean Systems research OCS, Royal Netherlands Institute for Sea Research (NIOZ), Texel, the Netherlands

*Correspondence to*: Frida S. Hoem (Frida.snho@gmail.com)

**Abstract.** At present, a strong latitudinal sea surface temperature (SST) gradient of ~16°C exists across the Southern Ocean, maintained by the Antarctic Circumpolar Current (ACC) and a set of complex frontal systems. Together with the Antarctic ice masses, this system has formed one of the most important global climate regulators. The timing of the onset of the ACC-system, its development towards modern-day strength, and the consequences for e.g., the latitudinal SST gradient around the southern Atlantic Ocean, are still uncertain. Here we present new TEX$_{86}$-biomarker records, calibrated to SST, from two sites located east of Drake Passage (southern South Atlantic) to assist in better understanding two critical time intervals of prominent climate transitions during the Cenozoic: The Late Eocene–Early Oligocene (ODP Site 696) and Middle–Late Miocene (IODP Site U1536) transitions. Our results overall show rather temperate conditions (20–11°C) during the Late Eocene to Early Oligocene interval, with a weaker latitudinal SST gradient (~8°C) across the Atlantic sector of the Southern Ocean compared to present day (~16°C). We ascribe the regional similarity in SSTs across the Late Eocene–Early Oligocene South Atlantic to a persistent, strong Subpolar Gyre circulation, connecting all sites, which can only exist in absence of a strong throughflow across the Drake Passage. Surprisingly, the southern South Atlantic records show comparable SSTs (~12–14°C) during both the Earliest Oligocene Oxygen Isotope Step (EOIS, ~33.65 Ma) and the Miocene Climate Optimum (MCO, ~16.5 Ma). Apparently, maximum Oligocene Antarctic ice volume could coexist with warm ice-proximal surface ocean conditions, while at similar ocean temperatures, the Middle Miocene Antarctic ice sheet was strongly reduced. Southern South Atlantic SSTs cooled to ~5°C at the onset of the Middle Miocene Climate Transition (MMCT, 14 Ma), making it the coldest oceanic region recorded around Antarctica and the likely main location for deep water formation. The already cold southern South Atlantic conditions at MMCT meant it experienced little cooling during the latter part of the Miocene, which contrasts the profound cooling due to northward expansion of the Southern Ocean frontal systems in the lower latitudes and other sectors of the Southern Ocean.



## 1 Introduction

Today, Southern Ocean surface flow is dominated by the strongest ocean surface current on Earth, the Antarctic Circumpolar Current (ACC). This wind-driven, eastward flowing surface current is associated with strong (~16°C, between ~45–60°S) meridional gradients in oceanographic conditions and temperature (Locarnini et al., 2018). Questions remain about the timing

and nature of the development of the ACC and concomitant evolution the complex Southern Ocean frontal systems (Fig. 1). A primary prerequisite for the existence of a strong ACC is an unobstructed latitudinal band of (deep ocean) water (Orsi et al., 1995; Barker and Thomas, 2004; Toggweiler et al., 2006), which is largely determined by the tectonic evolution and opening of the Tasmanian Gateway as well as the Drake Passage (Huber et al., 2004).

The tectonic evolution of the Tasmanian Gateway is relatively well constrained; early southern opening of the Tasmanian Gateway started around 49–50 Ma (Huber et al., 2004; Stickley et al., 2004; Bijl et al., 2013) with a change in course of tectonic drift of Australia from the Northeast to the North (Whittaker et al., 2007). Final breakup between Australia and Antarctica started around ~35.5 Ma, with ocean crust formation between southwestern Tasmania and Wilkes Land, Antarctica, and onset of bottom-water currents around 35.5–33.5 Ma (Stickley et al., 2004; Houben et al., 2019), although the strength of this so-

called 'proto-ACC' during the Oligocene remains debated. New field data reconstructing sea surface temperatures (SST) and water properties (Bijl et al., 2018; Hartman et al., 2018; Salabarnada et al., 2018; Evangelinos et al., 2020, 2022; Sauermilch et al., 2021; Hoem et al., 2021a, b, 2022; Hou et al., 2022) allow tracing the migration of frontal systems, which may be reconducted to opening of gateways. Furthermore, our understanding of the effect of the opening of the Southern Ocean gateways on the Australian-Antarctic Gulf oceanographic conditions has been improved through modelling exercises (England

et al., 2017; O'Brien et al., 2020; Sauermilch et al., 2021; Nooteboom et al., 2022). A recent SST compilation from around the Tasmanian Gateway (Hoem et al., 2022) shows that the latitudinal SST gradient between the subtropical front and the Antarctic Margin progressively increased from ~26 Ma onwards, due to cooling at Antarctic-proximal sites. Since the Tasmanian Gateway was already open, wide, and deep, this Antarctic cooling may have been related to the onset of deep ocean connections through the Drake Passage (Lyle et al., 2007; van de Lagemaat et al., 2021). The plate tectonic configuration of the Drake

Passage-Scotia Sea region is more complex than that of the Tasmanian Gateway, with various oceanic basins that opened at different times, and the separation of continental fragments from southernmost South America and from the Antarctic Peninsula (Kennett, 1977; Barker and Thomas, 2004; Maldonado et al., 2006; Lagabrielle et al., 2009; Pérez et al., 2019; 2021). The timing and nature of the opening, widening, and deepening of the Drake Passage has also been much debated, and placed between 50 Ma and 16 Ma (Barker et al., 2007; Livermore et al., 2007; Eagles and Jokat, 2014; Maldonado et al., 2014; Pérez

et al., 2021; van de Lagemaat et al., 2021). While some deep ocean sedimentary records suggest oceanographic rearrangements that were possibly linked to an early Drake Passage opening (Kennett, 1977; Scher and Martin, 2006; López-Quirós et al., 2019; López-Quirós et al., 2021), we lack knowledge of the long-term history of the evolution of SST gradients in the South



Atlantic. South Atlantic SST gradient development could be used to interpret phases of Antarctic cooling, strengthening of the ACC and shifts in the frontal systems in turn linked to the throughflow of surface and deep waters through the Drake Passage.


Recent drilling efforts in and around the Scotia Sea during International Ocean Discovery Program (IODP) Expedition 382 (Weber et al., 2021a), and a revisit of previously drilled records in the Weddell Sea during ODP Leg 113 offer improved spatial coverage of sedimentary records across two prominent climate transitions, viz (1) the Eocene-Oligocene transition (EOT; 33.7 Ma) and (2) the transition from the Miocene Climate Optimum (MCO,~16.5 Ma) to the mid-Miocene Climate Transition

(MMCT, ~14.7–13.8 Ma) and the late Miocene Cooling (LMC; ~7–5.4 Ma). Both transitions are marked by increases in benthic foraminiferal $\delta^{18}O$, suggesting cooling and expansion of the Antarctic ice sheet (Rohling et al., 2022). Given the paucity of carbonaceous sediments in this region, typically employed for paleotemperature reconstructions, we here choose to generate lipid biomarker (TetraEther indeX of tetraethers with 86 carbon atoms (TEX$_{86}$)) proxy SST reconstructions, based on isoprenoid glycerol dialkyl glycerol tetraether (isoGDGT) distributions, from the northern Weddell Sea at ODP Site 696 (Late

Eocene–Early Oligocene) and southern Scotia Sea IODP Site U1536 (mid–late Miocene) (Fig. 1). We compare our findings to available TEX$_{86}$, alkenone unsaturation index (U$^{k'}_{37}$) and clumped isotope ($\Delta_{47}$) derived SST records from the South Atlantic region (Fig. 1) for a reconstruction of paleoceanographic conditions.





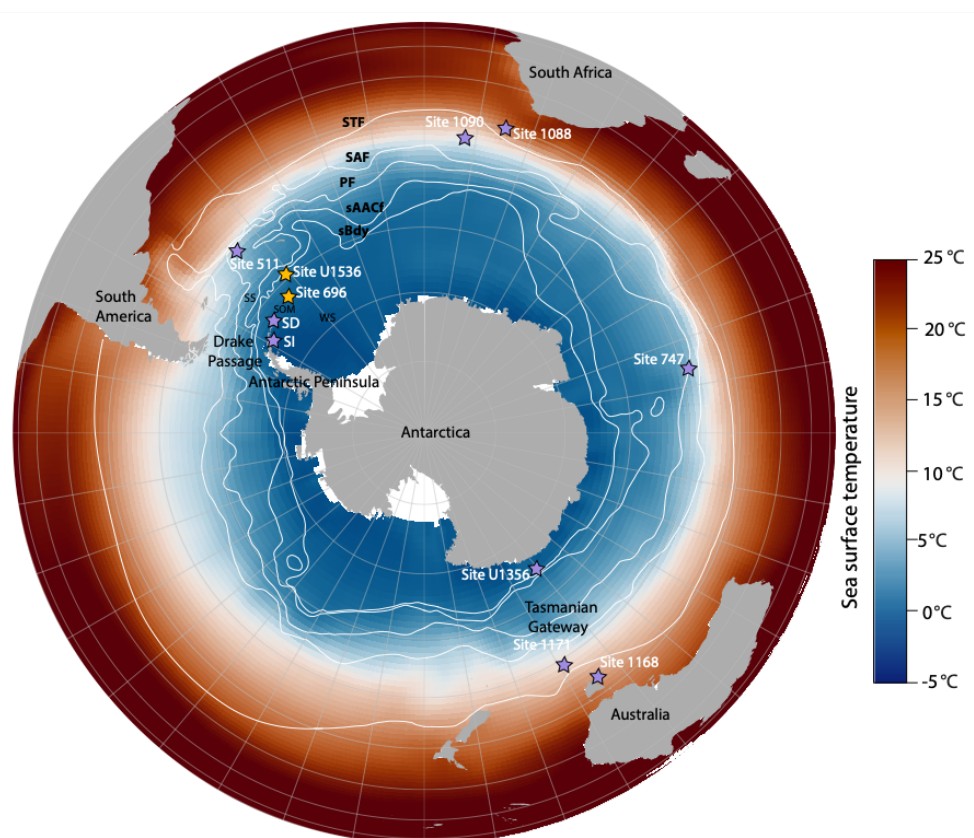


**Figure 1. Present day map of the Southern Ocean showing the location of the drill sites used in this study. Grey areas represent present-day land masses. The colors show average Summer (January) SSTs from 1971-2000 (Reynolds et al., 2002). The white lines represent the smoothed, simplified position of circumpolar fronts interpreted by Orsi et al. (1995). From north to south: The Subtropical front (STF); the Subantarctic front (SAF); the Polar Front (PF); the southern ACC Front (sACCf); and the Southern Boundary (SBdy) front. SI=Seymour Island, SD=SHALLDRILL, WS= Weddell Sea, SS=Scotia Sea.**


## 2 Material

## 2.1 Sedimentary drill cores

### 2.1.1 Site 696 lithology, age model and depositional setting

ODP Leg 113 Site 696 was drilled in the northern Weddell Sea (61°50.959'S: 42°55.996'W), on the South Orkney

Microcontinent (SOM; Fig. 1), located south of the Southern Boundary (SBdy) front. The site had a late Eocene paleolatitude

of ~67°S (paleolatitude.org Version 2.1; van Hinsbergen et al. (2015), using the paleomagnetic reference frame of Torsvik et

al. (2012), and the geological reconstruction of Seton et al. (2012)).



Lithological descriptions and age constraints are gathered from López-Quirós et al. (2021; Fig. 2, Supplementary Table 1).
The age model is based primarily on calcareous nannofossil biostratigraphy (Wei and Wise Jr, 1990; Villa et al., 2008), and updated age constraints from organic walled dinoflagellate cysts (dinocysts) published previously (Houben et al., 2013; Houben et al., 2019; López-Quirós et al., 2021), which places the studied section of 607.6–548.9 mbsf (Cores 59-53 at 36.0 to 33.2 Ma (Houben et al., 2013, Supplementary Table 1). In the sediments overlying the lower Oligocene interval, 532–529.8 mbsf Cores 52-51), no specific age constraint was determined, but dinocyst analysis indicate Oligocene age. In Core 50, age

deterministic diatoms (*Denticulopsis maccolummii and Actinocyclus ingens*) were found and dated to the middle Miocene (17.6 - 15.4 Ma; Carter et al., 2017 or more accurate: 16.7–16.5 Ma; López-Quirós et al. .in prep).





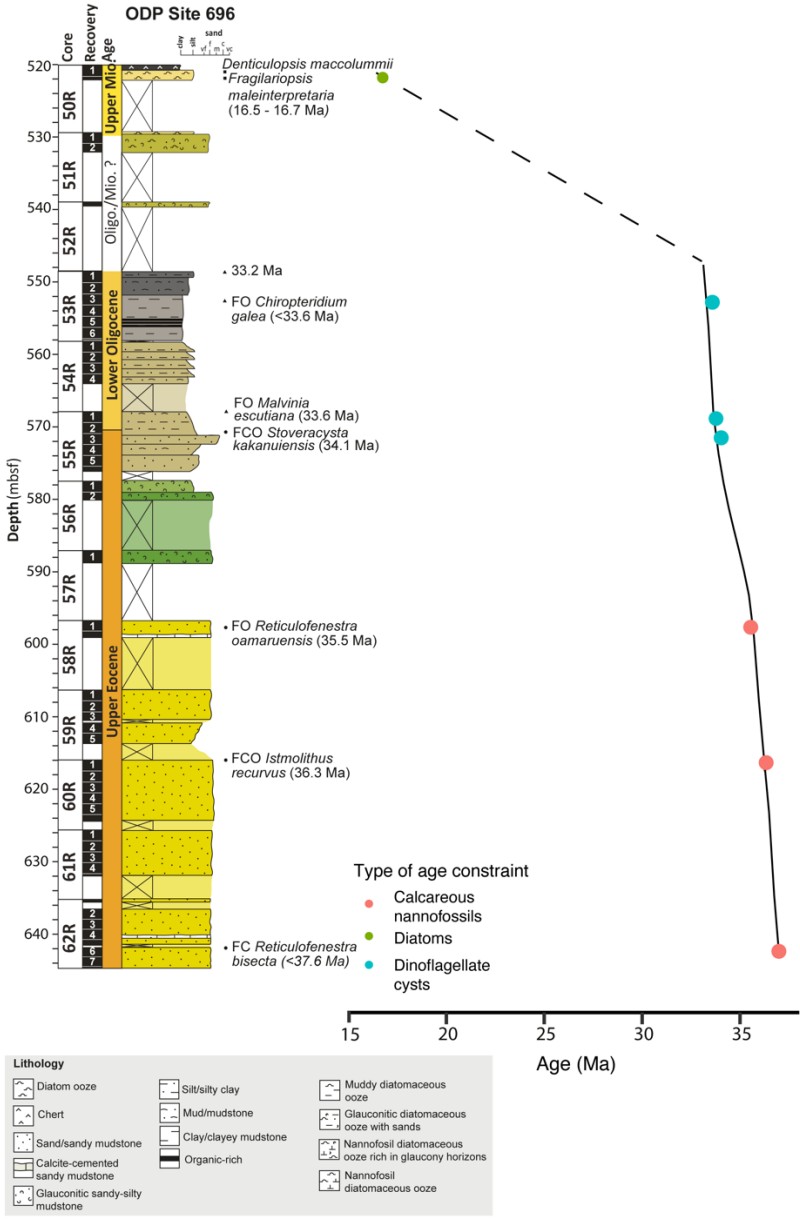

**Figure 2. Age-depth model of ODP Site 696 (Supplementary Table 1) and lithological log modified after López-Quirós et al. (2019);**
**(2020); (2021) and Barker et al. (1988), including new constraints of the uppermost interval (Core 50) from López-Quirós et al. (in prep.).**

The studied sediment package at Site 696 (607.6–521.08 mbsf) consists of; 1) organic-rich sandy mudstone facies (607.6–606.9 mbsf), 2) glaucony-bearing packstone facies (606.9 to 569.7 mbsf), 3) claystone and limestone facies (569.7 to 548.9 115 mbsf), 4) rhythmically interbedded sandy mudstone facies with glauconite bearing sandstone beds (548.9 to 529.8 mbsf), and



5) pelagic sediments, predominately biosiliceous diatom ooze (522–521 mbsf) (López-Quirós et al., 2019; 2020; 2021; in prep.). The glauconitic packstone beds of latest Eocene age (606.9 to 569.7 mbsf, ~35.5–34.1 Ma) are attributed to a decline in terrigenous input to Site 696 and increased winnowing (López-Quirós et al., 2019). The opening of the Powell Basin (at the tip of the Antarctic Peninsula) caused subsidence of the South Orkney Microcontinent, thereby decreasing the delivery of

terrigenous sediments to the site (López-Quirós et al., 2019). The Antarctic-proximal location makes the site prone to influx of terrigenous material transported by icebergs. Indeed, López-Quirós et al. (2021) show unequivocal evidence for the oldest iceberg rafted debris (IRD) deposited across the EOT (34.1–33.6 Ma, 576–568 mbsf) at Site 696. Across the EOT sediments reflect more distal and deeper environments while the SOM continued to deepen (López-Quirós et al., 2021). Eutrophic surface water conditions, indicative of productive and somewhat shallow-water and reduced-oxygen conditions are indicated by

dinocysts (Houben et al., 2019) and sedimentary facies (López-Quirós et al., 2019; 2021). The dominance of large sized heterotrophic Protoperidiniacean dinocysts in the earliest Oligocene is suggested to reflect seasonal sea-ice conditions (Houben et al., 2013). During the earliest Oligocene (~33.6–33.2 Ma), the SOM shelf subsided, and biological production increased, partially driven by upwelling along the shelf (López-Quirós et al., 2021). Deposition of moderately to intensely bioturbated silty-mudstones during the EOT is attributed to the continued subsidence-related deepening at Site 696 (López-Quirós et al.,

2021). Above the claystone/clayey mudstones of the lowermost Oligocene (Core 53R, Fig. 2) we find rhythmically interbedded sandy mudstones with glauconite-bearing sandstone beds, a result of reworked sediments, deposited under bottom current activity and possibly slumping (López-Quirós et al., 2020). The nature of this sediment complicates dating of this material. There is likely a break in the sedimentation (hiatus) around 529 mbsf (the top part of Core 51R), with a sharp contact from glauconitic-bearing sandstone to mud bearing diatom ooze 522–521 mbsf (Core 50R), dated to ~16.7–16.5 Ma (López-Quirós

et al. in prep).

### 2.1.2    Site U1536: lithology, age model and depositional setting

Site U1536 is located in Dove Basin, in the southern Scotia Sea (59°26.4608'S, 41°3.6399'W, 3220 m water depth). The site was drilled to study the Neogene flux of icebergs through "Iceberg Alley", the main pathway along which icebergs calved from the margin of the Antarctic ice sheet drift into the warmer waters of the ACC (Weber et al., 2021a). Today, the site is

located just south of the southern ACC Front (sACCf) and the SBdy front and is seasonally covered by sea ice (Fig. 1). The rotary drilling at Hole U1536E penetrated down to 643 mbsf. Sediments have moderate to high core disturbance and biscuiting or brecciated core material due to the rough nature of rotary drilling and compaction of gravel rich material. The lithology of the studied interval from Hole U1536E 640–450 mbsf (Cores 33R–13R) consists of silty clays with interbedded diatom ooze (Fig. 3), with an estimated average sedimentation rate of ~4.3 cm/kyr (3.9–6.4 cm/kyr; Pérez et al., 2021). The shipboard bio-

and magnetostratigraphic age model was used to date the sediments (Weber et al., 2021b, Supplementary Table 2). Sediments between 480–450 mbsf (Cores 16R–13R) have an age of 6–5 Ma based on bio- and magnetostratigraphy. Diatom biostratigraphy between 548–535 mbsf (Cores 24R–22R) indicate ages of 7.7–6.4 Ma. The sediment directly overlying Reflector-c (Weber et al., 2021b) at 617–570 mbsf (Cores 30R–26R) have an age of 8.4 Ma (Pérez et al., 2021). The sediments



below Reflector-c, at 622 mbsf (Core 31R), are dated to ~14.2 Ma, and as the lithologic contact is not recovered, it means that
Reflector-c could represent a prolonged time interval of slow sedimentation rates or non-deposition or erosion. The sparse
brecciated lithology fragments in the lower cores below the Reflector-c (566 mbsf), consists of lithified mudstone and gravel-
conglomerate-breccia (Weber et al., 2021b; Perez et al., 2021).

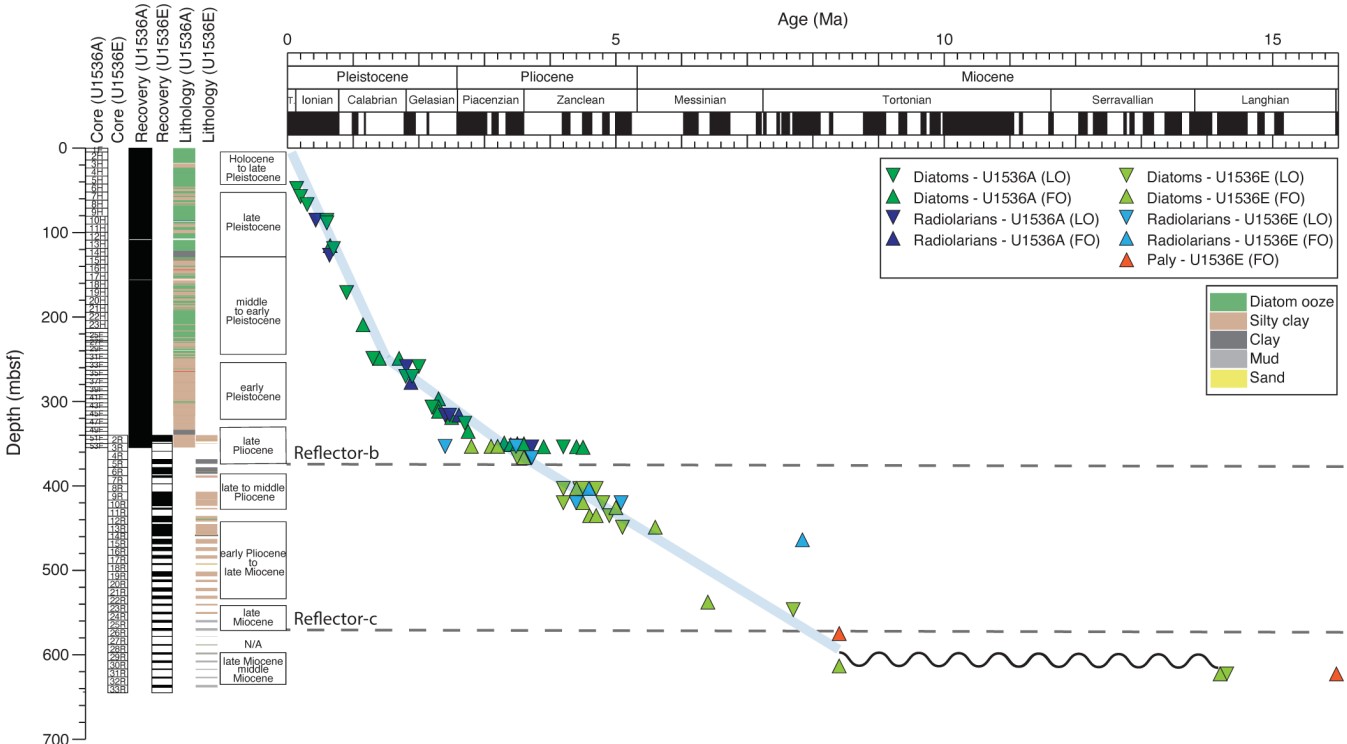

**Figure 3. Site U1536 age model and lithology modified after the IODP Expedition 382 shipboard report (Weber et al., 2021b). The
depth and age of the stratigraphic discontinuities (seismic reflectors) are derived from Perez et al. (2021) (Supplementary Table 2,
Perez et al., 2021). LO = Last occurrence, FO = First occurrence.**

## 3      Methods

### 3.1      Lipid Extraction and glycerol dialkyl glycerol tetraether (GDGT) analysis

Lipid extraction of sediments from Site 696 was performed at the Laboratoire d'Océanographie et du Climat, Expérimentations
et Approches Numériques (LOCEAN-Sorbonne Université, Paris, France). First, 71 sediment samples were freeze-dried and
crushed to a fine powder. Total lipids were extracted from ~9.5 to 15 g of homogenized sediment using a solvent mixture of
40 ml DCM:MeOH (3:1, v/v). The polar fraction was eluted from the total lipid extracts (TLEs) using 3 ml DCM:MeOH (1:1,
v/v) over a silica column. The aploar fraction was separated from the polar lipids over a silica column using 3 ml hexane as
eluent. The polar lipid fractions were sent to Utrecht University for further analysis. Sediment samples from Site U1536E were



processed for GDGT analysis by lipid extraction at Utrecht University, from 10 g freeze-dried and manually powdered sediments using a Milestone Ethos X microwave system and adding dichloromethane:methanol (DCM:MeOH; 9:1, v/v). The TLEs were first filtered through a NaSO$_4$ column to remove potential remaining water and sediments. The TLEs were then separated on an activated Al$_2$O$_3$ column into apolar, ketone and polar fractions, using hexane:dichloromethane (DCM) (9:1,

v/v), hexane:DCM (1:1, v/v), and DCM: MeOH (1:1, v:v) as eluents, respectively. All polar fractions were dried under N$_2$. A known amount of C$_{46}$ GTGT standard was added to the polar fractions from Sites 696 and U1536, which was subsequently dissolved in hexane:isopropanol (99:1, v/v) to a concentration of ~2 mg ml$^{-1}$ and filtered through a 0.45 µm polytetrafluorethylene filter. After that, the dissolved polar fractions were injected and analyzed by ultra-high performance liquid chromatography/mass spectrometry (UHPLC/MS) according to the method described by Hopmans et al. (2016), using

an Agilent 1260 Infinity UHPLC system coupled to an Agilent 6130 single quadrupole mass detector, at Utrecht University. Selected ion monitoring (SIM) was used to identify the GDGTs using their [M+H]$^+$ ions and integrated using ChemStation software. Samples with very low concentrations (i.e., peak area <3,000 mV/s and/or peak height <3× background signal) of any GDGT included in TEX$_{86}$ were excluded from analysis.

### 3.2    GDGT indices for non-thermal overprints on TEX$_{86}$

The TEX$_{86}$ SST proxy is based on the temperature dependence of the number of cyclopentane rings in GDGT membrane lipids produced by marine Thaumarchaeota and calculated as defined by Schouten et al. (2002):

$$TEX_{86} = ([GDGT\text{-}2]+[GDGT\text{-}3]+[cren'])/([GDGT\text{-}1]+[GDGT\text{-}2]+[GDGT\text{-}3]+[cren'] \qquad (1)$$

The use of TEX$_{86}$ as a proxy for SST relies upon the assumption that isoprenoidal GDGTs in marine sediments are principally derived from membrane lipids of marine pelagic Thaumarchaeota (Schouten et al., 2013). However, in some environments, non-thermal factors may alter the distribution of isoprenoidal GDGTs stored in the sediment and thus the temperature signal (Supplementary information). We assess potential non-thermal effects on GDGT distributions prior to translating TEX$_{86}$ into SSTs. We use the branched and isoprenoid tetraether (BIT) index to assess possible overprints of terrestrial GDGT input

(Hopmans et al., 2004), the fractional abundance of the crenarchaeol isomer over that of crenarchaeol (fcren'; O'Brien et al., 2017), the methane index to identify contributions of methanotrophic archaea (MI; Zhang et al., 2011) or methanogenic archaea using the ratio of GDGT-0/crenarchaeol (Blaga et al., 2009), the GDGT-2/GDGT-3 ratio to assess contributions of a deep dwelling GDGT producing community (Taylor et al., 2013) and the ΔRing Index to identify GDGT distributions that deviate from modern analogues (ΔRI; Zhang et al., 2016) (Fig. S1C, panel 2–8 and Fig. S2C, panel 2–8).






### 3.3 TEX$_{86}$ calibration

The empirical relationship between TEX$_{86}$ values and SST has appeared to be not always straightforward, as reflected by continued revisions of the approach of TEX$_{86}$ – SST calibrations (Kim et al., 2010; Tierney and Tingley, 2015; Ho and Laepple, 2016; O'Brien et al., 2017; Dunkley Jones et al., 2020). Particularly, the relationship between TEX$_{86}$ and SST seems to become obscured in both extreme ends of the core-top calibration: at or above modern SSTs, and in cold polar regions. However, for the time intervals (Late Eocene–Early Oligocene and Middle–Late Miocene) and locations we are targeting, we expect SSTs within the intermediate temperature range. In this study, we used the regionally varying BAYSPAR SST calibration of Tierney and Tingley (2015) to reconstruct SST (±4°C standard calibration error) from TEX$_{86}$ values. BAYSPAR compares measured TEX$_{86}$ values to modern TEX$_{86}$ values obtained from surface sediments, to derive linear regression parameters, and propagates uncertainties in the surface sediment data into resulting temperature predictions (Tierney and Tingley, 2015). Even when GDGT-2/GDGT-3 ratios indicate that deeper dwelling GDGT producers do not contribute to the sedimentary signal, the GDGTs could still originate from the subsurface (50–200 m water depth) rather than the sea surface (Schouten et al., 2013). However, since the BAYSPAR calibration translates TEX$_{86}$ values to SSTs, we will, therefore, refer to the proxy results as such in the remainder of this work. We used a standard deviation of ±20°C and a prior mean of 20°C for the Late Eocene–Early Oligocene interval and 15°C for the Miocene. We also compare the BAYSPAR-derived SST estimates with those based on the exponential function from Kim et al. (2010) and the linear function by O'Brien et al. (2017). The SST records all show similar trends, but BAYSPAR-derived SSTs are usually cooler compared to those obtained from the functions of Kim et al. (2010) and O'Brien et al. (2017) (Supplementary Table 3, 4, Fig. S3, S4).

### 4 Results

### 4.1 Site 696 GDGT distributions and TEX$_{86}$-SST trends

A total of 71 samples from ODP Site 696 were analysed for TEX$_{86}$ palaeothermometry (Supplementary Table 3). The GDGT pool consist of 90 ± 5% isoGDGTs and 10 ± 5% brGDGTs, resulting in BIT index values <0.2 (Fig.5 b). The isoGDGT distributions indicate that GDGTs are primarily produced by surface ocean-dwelling Thaumarchaeota (Supplementary information, Fig. S1). In total, 9 samples had higher ΔRI values than the cut off of 0.3 (Fig. S1B) indicating a potential non-thermal overprint on the GDGT distribution and are excluded from SST analysis (triangles in Fig. 4). The TEX$_{86}$-based SST record from Site 696 (607–521 mbsf, 36–33 Ma and ~16.5 Ma, Fig. 4) thus consists of 62 data points, and predominantly ranges between 12°C and 18°C, although the total temperature range is from 4°C to 25°C (± 4°C standard error). There is a general cooling trend of ~8°C between the Upper Eocene and the Lower Oligocene (from 20°C to 12°C between 610 and 558 mbsf) interval, but with high sample-to-sample variability. This is followed by an average ~5°C increase in temperature between ~558 and ~552 mbsf. In the organic rich interval at ~555 mbsf (Core 53R, section 5, 33.5–33.2 Ma), 35 sediments were analyzed at high resolution (every 3 cm, ~15 kyrs resolution). The TEX$_{86}$-SST record shows high-amplitude variability



(total range; 4–21°C, with most datapoints falling within the range between 8–17°C). At ~550 mbsf (Lower Oligocene) SSTs rapidly decreases to 10°C. In the Middle Miocene (~520 mbsf, n=2) SSTs values are ~14°C.



**Figure 4. a)** TEX$_{86}$-SST data (BAYSPAR calibration; Tierney and Tingley, 2015). Datapoints marked by a triangle represent samples (n=9) with a potential non-thermal overprint on the GDGT distribution (outliers) (Supplementary information), **b)** BIT-index values (Hopmans et al., 2004) plotted next to the lithology and age constraints of Site 696, which are modified after Lopez-Quiros et al. (2021), based on Barker et al. (1988) and Lopez-Quiros et al. (2019, 2020), including new constraints of the uppermost interval (Core 50) from López-Quirós et al. (in prep.).





## 4.2     Site U1536 GDGT distributions and TEX$_{86}$-SST trends

A total of 40 sediment samples from IODP Hole U1536E were processed for GDGT analysis (Supplementary Table 4), of which 14 had GDGT concentrations below detection limit (Fig. 5). The Site U1536 GDGT pool consists of variable amounts of both isoGDGTs and brGDGTs, where brGDGTs are relatively more abundant in the middle (500–560 mbsf) and top parts of the record (450 mbsf) (Fig. S2A), resulting in high BIT-index values in these middle and top intervals of the record (Fig. 5b). GDGT distributions in 12 sediment samples were outside the range of what is considered reliable for multiple indicator proxies (triangles in Fig. 5a, Supplementary information, Fig. S2). For the remaining 14 samples TEX$_{86}$ values were translated into SSTs using the BAYSPAR calibration (Fig. 5a). The obtained record shows temperatures of 5–11°C for the Middle Miocene (619–640 mbsf) and 1.5–5°C for the Upper Miocene (570–450 mbsf).

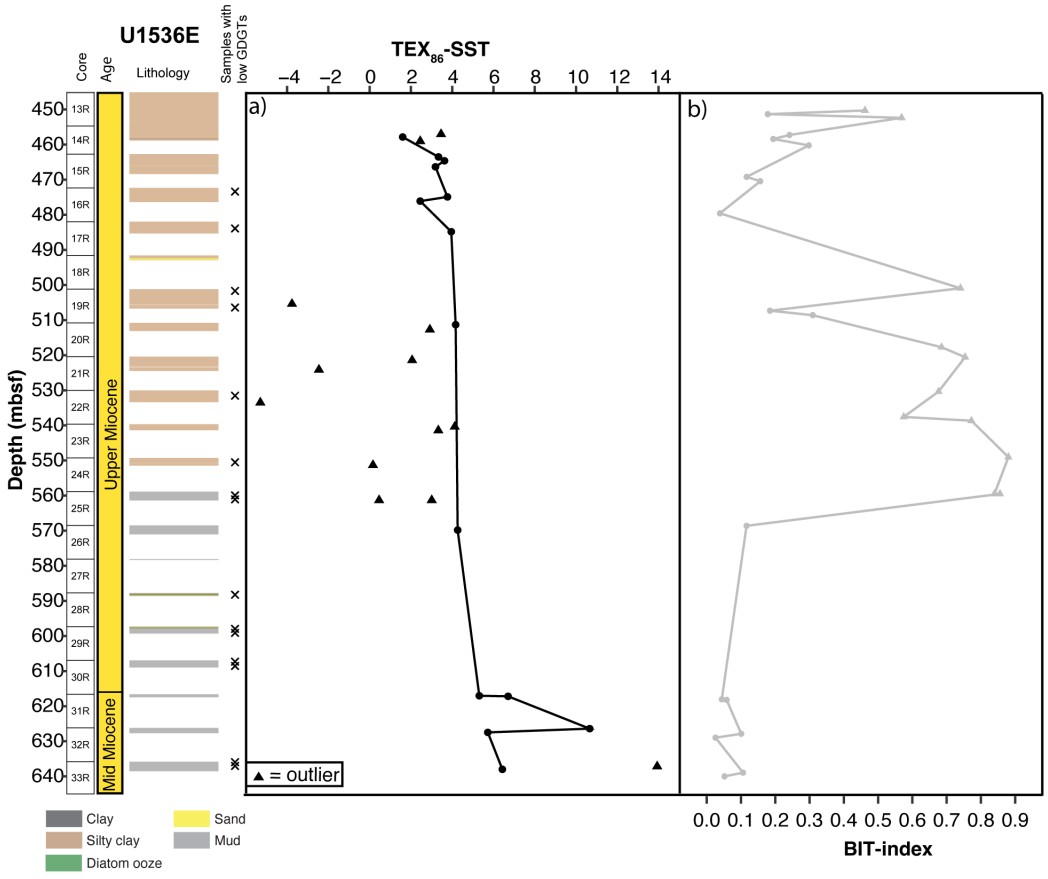

**Figure 5.**
**Biomarker indices from Site U1536. Left panel shows core recovery, estimated age and lithological composition (Weber et al., 2021a). The crosses (X) indicate samples with GDGT concentrations below detection threshold. a) TEX$_{86}$-SST (BAYSPAR calibration; Tierney and Tingley, 2015). Triangles mark samples with a potential non-thermal overprint on the GDGT distribution (outliers) (Supplementary information), b) BIT-index (Hopmans et al., 2004) values.**



## 5    Discussion

### 5.1    Late Eocene-Early Oligocene South Atlantic SST conditions

We have compiled available South Atlantic SST records from Site 1090 (Lie et al., 2009), Site 511 (Houben et al., 2019) and Seymour Island (Douglas et al., 2014), with our new SST record from Site 696 (Fig. 6) to put the SST records into a broader regional context and discuss the surface oceanographic development during the Late Eocene–Early Oligocene. Our SST record for Site 696 (yellow in Fig. 6A) shows warm-temperate conditions (SST range: 22–14°C) during the latest Eocene (~36.5–

33.6 Ma) and on average decreasing SSTs (~15–9°C) in the earliest Oligocene (33.6–33.2 Ma). Miospores at Site 696, believed to be of local origin from the South Orkney Microcontinent, changed concomitantly from southern beech, *Nothofagus*-dominated vegetation to an increase in gymnosperms and cryptogams, accompanied by a rapid rise in taxon diversity after the Earliest Oligocene Oxygen Isotope Step (EOIS, ~33.65 Ma, 568.82 mbsf) (Thompson et al., 2022). This shift in vegetation to a cooler and dryer climate occurred after the onset of earliest glacial expansions (~34.1 Ma) and corresponds to our SST drop

below 12°C and the more common IRD in the sediment at this depth (see also López-Quirós et al., 2021). Sedimentological investigations by López-Quirós et al. (2021) for the same interval showed deepening of the South Orkney Microcontinent shelf and enhancement of biological production, possibly due to upwelling along the shelf, leading to low oxygen conditions at the seafloor. The high-amplitude SST variability in our $TEX_{86}$-SST record (~4–8°C) would suggest that this upwelling regime was strongly variable. The variability in upwelling conditions could be induced by strong fluctuating ice sheet expansion and

retreat and shifts in wind patterns and ocean frontal systems.





**Figure 6. A.** The upper panel shows biomarker-based SST trends from this study (Site 696) compared with data from Site 1090 (Lie et al., 2009) and Site 511 (Houben et al., 2019) and clumped isotope-based SST from Seymour Island (Douglas et al., 2014). The lower record represents the global benthic foraminiferal δ18O compilation, smoothed by a locally weighted function over 20 kyr (black curve) (CENOGRID; Westerhold et al., 2020). The blue curve is the LOESS smoothed (span = 0.2). **B.** Reconstructed paleogeographic map at 34 Ma, based on the GPlates reconstruction of van de Lagemaat et al. (2021) in the paleomagnetic reference frame of Torsvik et al. (2012). Arrows show the ocean circulation derived from GCM circulation model by Goldner et al., 2014.

TEX$_{86}$- and U$^{k'}_{37}$-based SST reconstructions from ODP Site 511 differ slightly in trends across the EOT, with U$^{k'}_{37}$ showing amplified cooling compared to TEX$_{86}$. U$^{k'}_{37}$-temperature data reflect surface water temperature following the niche of the haptophyte algae that produce the alkenones on which the U$^{k'}_{37}$ is based, while the GDGTs that mostly contribute to the TEX$_{86}$ signal stored in the sediments are likely produced in the subsurface (50–200 m water depth) (Schouten et al., 2013). Although U$^{k'}_{37}$ is calibrated to mean annual temperature (Müller et al., 1998), in some settings it reflects a seasonally biased temperature





(e.g., Ternois et al., 1997). In fact, modern high-latitude alkenone producing haptophyte communities primarily bloom in early

spring (Herbert et al., 1998). Thus, at Site 511, the amplified cooling in $U^{k'}_{37}$ might reflect a shift towards early spring blooming, which could explain its amplified cooling trend across EOT, relative to that in $TEX_{86}$. As $TEX_{86}$-based SSTs are consistently higher than $U^{k'}_{37}$-based SSTs at this site, we argue that $TEX_{86}$ represents a warmer season than $U^{k'}_{37}$ and that it does seem to be reflecting a surface water signal rather than a subsurface signal.

The clumped isotope ($\Delta_{47}$)-based SST datapoint from Seymour Island (34 Ma, purple square Fig. 6) shows a similar temperature (13°C) to that derived from the $U^{k'}_{37}$ and $TEX_{86}$ proxies at the same site (Douglas et al., 2014). The correspondence of the biomarker-derived SSTs with that from Eurhomalea (bivalve) $\Delta_{47}$ (mean annual temperature), i.e., three completely independent temperature proxies, adds reliability to the temperature proxies to accurately reflect SST at this site. Interestingly, the clumped isotope temperature is very similar to SSTs from Sites 696 and 511, even though there was a paleolatitudinal

difference of ~12° between Site 511 and Seymour Island (Fig. 6). This suggests that all sites experienced a similar paleoceanographic regime. The subtropical Site 1090 is the warmest site (~19–27°C) in our compilation, which is expected given its lower paleolatitude. The absolute SSTs of Sites 696 and 511 are strikingly similar across the EOT, with Site 511 being only slightly warmer (by ~2°C) than Site 696 in the Late Eocene, and similar to Site 696 in the Early Oligocene (Fig. 6A). This results in a lack of an increase in SST gradient across the EOT, while today both sites are separated by the strong

ACC and an SST gradient of >7°C (Locarnini et al., 2018).

Houben et al. (2019) found widespread glauconite-bearing facies in Southern Ocean sediment cores from about 35.7 Ma and attributed this to invigoration of the basin-wide bottom waters, which led to sediment winnowing, higher biological productivity and aided cooling of surface waters around Antarctica. This process could have been strengthened by concomitant

Eocene Tasmanian Gateway widening and deepening (Sijp et al., 2014; 2016; Sauermilch et al., 2021) and/or invigorated atmospheric circulation induced by Late Eocene polar amplification of $CO_2$-forced cooling (Scher et al., 2014; Cramwinckel et al., 2018). Site 696 (~588.8 to 577.9 mbsf) also contains glaucony-bearing facies of comparable age, which López-Quirós et al. (2019) interpreted as the result of invigorating bottom currents. Stronger bottom currents were thought to be the result of the opening of the Powell Basin and the establishment of oceanic upwelling, which drove regional cooling and increased

primary productivity. While this could be a plausible explanation, the tectonic opening of Powell Basin did not result in an increase in the SST gradient between Sites 696 and 511. Alternatively, both sites were similarly affected by the opening of the Powell Basin and the resulting atmospheric and/or oceanographic changes. The constant SST gradient across the EOT would further suggest that both sites remained under the same oceanographic regime, implying that the Antarctic ice sheets only developed on land and therefore did not exert an influence on SSTs and that throughflow across the Drake Passage did not

change across the EOT. Tectonic evidence suggests that the Drake Passage was narrow, with little deep-water connection from the Pacific to the Atlantic around the EOT (Livermore et al., 2007; Eagles and Jokat, 2014; van de Lagemaat et al., 2021). Fully coupled climate models simulating oceanographic conditions across the EOT by Goldner et al. (2014) and Kennedy et



al. (2015) show that with a restricted Drake Passage, the Subpolar Gyre in the South Atlantic would expand further north in the absence of throughflow (Fig. 6B). Moreover, recent model experiments (Hill et al., 2013; England et al., 2017) also showed

that a closed Drake Passage could sustain warm currents to the Antarctic Margin. In these models, a Southern Ocean without open gateways featured wind-driven clockwise gyres in the South Pacific and South Indian/Atlantic Ocean basins (Huber et al., 2004; Sauermilch et al., 2021) (Fig. 6B) that would advect warm surface waters toward the Antarctic coast. Recent ocean model simulations show that this would lead to SSTs reaching 19°C in the Australian-Antarctic Basin and 15–17°C in the subpolar Pacific and Atlantic (Sauermilch et al., 2021). This is similar to our TEX$_{86}$-SST record, although the model results

are to some extent dependent on the radiative forcing imposed. We thus propose that the small SST difference between Sites 511 and 696 is the result of a restricted Drake Passage during the EOT – earliest Oligocene.

In summary, cooling of South Atlantic surface waters across EOT is in broad agreement with the average Southern Ocean-wide temperature drop (Kennedy-Asser et al., 2020), the increase in benthic foraminifer $\delta^{18}$O values as a result of deep sea

cooling, a drop in atmospheric $p$CO$_2$ levels across the EOT (https://www.paleo-co2.org; Pearson et al., 2009; Steinthorsdottir et al., 2016; Hoenisch, 2021) and the growth of a continent-wide Antarctic ice sheet (Bohaty et al., 2012). Step-wise cooling across the EOT (Hutchinson et al., 2021) is evident at Site 511 and in the U$^{k'}_{37}$ record at Site 1090. The first step occurs around 34.1 Ma and coincides with common IRDs at Site 696, suggesting glacial onset in the region (López-Quirós et al., 2021). The second step coincides with the Earliest Oligocene Oxygen Isotope Step (EOIS, 33.65 Ma; Hutchinson et al., 2021). Yet, the

lack of a strong SST gradient, and absence of a prominent increase thereof across EOT, suggest a persistent wind driven gyral circulation that connected all sites (Sauermilch et al., 2021). The position of the continents and the position of the winds had a strong effect on patterns of Southern Ocean circulation (Scher et al., 2015; Evangelinos et al., 2022), which during the EOT were not yet in line for a strong ACC. The southward shifts of westerly wind stress associated with the onset of Antarctic glaciation across the EOT (Kennedy et al., 2015) could expand the size of subtropical gyres and shrink the size of Subpolar

Gyres (Xing et al., 2022). However, an atmospheric forced invigoration of the Subpolar Gyre (Houben et al., 2019) did not induce stronger latitudinal temperature gradients between these sites according to our data.

### 5.2    Middle to Late Miocene

To investigate the cooling step in the mid-Miocene, the new Miocene SST records from the Antarctic-proximal southwest Atlantic Sites 696 and U1536 are compared to a record from the Antarctic Peninsula; SHALDRILL II (Core 5D; Tibbett et al.,

2022) and Wilkes Land (Site U1356, Sangiorgi et al., 2018). Due to the large age uncertainties in the sedimentary records of Site U1536 (Weber et al., 2021a; Perez et al., 2021) and SHALLDRILL II (Bohaty et al., 2011), we present the TEX$_{86}$-derived SSTs as average temperatures within two broad time intervals indicating the age uncertainties. We also compare SST trends from the above-mentioned sites to those from the subantarctic zone; ODP Site 1171, southwest Pacific Ocean (Leutert et al., 2020) and subtropical front; ODP Site 1088 (Herbert et al., 2018) in the Southeast Atlantic, and Site ODP 1168 west of



Tasmania (Hou et al., 2022). Further, we compare our new SST record to clumped isotope bottom water temperatures (BWT) from South Indian Ocean ODP Site 747 (Leutert et al., 2021) (Fig. 7).



**Figure 7. A. TEX₈₆-based SST data from Site 696 and U1536 (this study) compared to Southern Ocean wide SST and BWT records. Data of Site U1536 is displayed as two bar plots (red) showing the temperature ranges for the Middle Miocene (16–14 Ma) and late**
**Miocene (7–5.3 Ma), individual data points are shown as red dots. We compare our data to SST records from Wilkes Land Site U1356 (Sangiorgi et al., 2018), SHALDRIL II Core 5D (Tibbett et al., 2022; grey bars indicate the age uncertainty), Site 1168 (west of Tasmania; Hou et al., 2022), Site 1171 (southwest Pacific Ocean; (Leutert et al., 2020), U^{k'}₃₇- SST data from Site 1088 (Herbert et al., 2016) and clumped isotope bottom water temperature (BWT) data from Leutert et al. (2021) (Site 747). The black line is the benthic foraminiferal δ¹⁸O compilation, smoothed by a locally weighted function over 20 kyr (thin blue curve) (CENOGRID;**



**Westerhold et al., 2020). Thick blue curve is the LOESS smoothed (span = 0.2). The stippled vertical line indicates the age for the paleogeographic map below. B. Paleogeographic reconstruction at 16 Ma, based on the GPlates reconstruction of van de Lagemaat et al. (2021) in the paleomagnetic reference frame of Torsvik et al. (2012). Dashed black line represents the Miocene surface ocean currents derived from Herold et al. (2012). For the location of all sites in the data compilation in A see Fig. 1.**

The data compilation for South Atlantic SSTs (Fig. 7A) shows a 7°C cooling between Site 696 (yellow dots, ~14°±4°C, n=2)

during the Miocene Climate Optimum (MCO, ~16.5 Ma) and Site U1536 (red, ~7±4°C, n=5) in the Middle Miocene Climate Transition (MMCT, ~14.7–13.8 Ma). Both sites were located at comparable paleolatitudes during the Miocene (Fig. 7B). SSTs at Wilkes Land Site U1356 were warmer (17°C around 17 Ma) than at Site 696 (~14°C), even though Site U1356 was situated at a higher latitude (59°S) than Sites 696 and U1536 (~65°S). Additionally, a less pronounced cooling (SST ~16–12°C) occurred across the MMCT at Site U1356 (Sangiorgi et al., 2018), than the on average ~7°C cooling that our South Atlantic

low-resolution dataset shows. Thus, the South Atlantic was colder at the onset of MMCT than the Wilkes Land Antarctic margin. There was a ~6–10°C temperature difference between the South Atlantic Antarctic proximal Sites 696/U1536 and the subantarctic Site 1171 (southwest Pacific Ocean), with Site 1171 clumped isotope ($\Delta_{47}$) SSTs of 14–12°C and TEX$_{86}$ SSTs of 18–13°C during MMCT (Leutert et al., 2020). This suggests a relatively strong SST gradient between the coldest Antarctic-proximal regions and the subantarctic zone in the Middle Miocene, even though the subantarctic zone was likely at lower

paleolatitudes in the Australian-Antarctic Gulf than in the South Atlantic due to the more southerly position of Australia. The increase in temperature gradient in the complied SST records (Fig. 7A) in the Middle Miocene indicates breakdown of dominant gyral circulation since the early Oligocene and a strengthening of the ACC during the Middle–Late Miocene. The alkenone-based SST reconstructions at Site 1088 and TEX$_{86}$-based SST reconstructions at Site 1168, together representing subtropical front conditions, show MCO temperatures between 32–27°C that progressively cooled during the MMCT (~24–

14°C) (Hou et al., 2022) albeit less than at the Antarctic-proximal sites, which indicate an amplification of cooling at high latitudes. The clumped isotope ($\Delta_{47}$) BWT record from Site 747 in the South Indian Ocean (Leutert et al., 2021) shows strikingly similar temperatures to the reconstructed SSTs at Site U1536, which may suggest that the Weddell Gyre in the southern South Atlantic was an important region of deep water formation in the Miocene, like today (e.g., Orsi et al., 1999), and is in line with what modelling studies suggest for the Miocene (e.g., Herold et al., 2012). The 7°C southern South Atlantic

cooling between the MCO and MMCT occurs in a time of declining $p$CO$_2$ (Foster et al., 2012; Greenop et al., 2014) (Fig. 8D) and increasing benthic foraminiferal $\delta^{18}$O (Fig. 7A, 8C), reflecting an increasingly colder climate with larger, more permanent ice sheets on the Antarctic continent (Lewis et al., 2008; Shevenell et al., 2008; Holbourn et al., 2018; Levy et al., 2019). We therefore infer that in the Middle Miocene, the southern South Atlantic records at Sites 696 and U1536 respond sensitively to the $p$CO$_2$-induced cooling.


The (late) middle Miocene (12.8–11.7 Ma) TEX$_{86}$ data from SHALDRILL II Hole 5D (Tibbett et al., 2022) indicate that relatively warm high latitude SSTs (mean: 10.2°± 3.4°C) persisted towards the Late Miocene. By the latest Miocene (~6.4–5.2 Ma), temperatures at Site U1536 cooled to a mean SST of 3.4°C (n=8) which match those at SHALLDRILL II in the earliest Pliocene (5.1–4.3 Ma, mean 2.8°C, n=3). The 3–5°C SST decrease at Site U1536 between the MMCT (~14 Ma) and



the latest Miocene (~6.4–5.2 Ma) is perhaps surprisingly small compared to the ~10°C cooling at the subtropical front sites at the same time. We surmise that this high-latitude cooling was subdued in this time interval because southern South Atlantic surface waters (Site U1536) were already cold during the MMCT (Fig. 7A). It is the rest of the Southern Ocean that experienced pronounced cooling in this time interval. Based on the Southern Ocean records (Fig. 7A), the South Atlantic record the coldest SSTs and was therefore possibly the coldest region of the Southern Ocean already since at least the Middle Miocene. However,

there is a strong lack of Antarctic proximal records (e.g., no records from the Ross Sea; Levy et al., 2019) that cover the MMCT to Late Miocene, albeit partly due to glacial advances, to draw a full picture of circum-Antarctic cooling since the MMCT.

Both Sites U1536 and 747 show a lack of latest-MMCT – Late Miocene cooling in SST and BWT, respectively (Leutert et al., 2021). Leutert et al. (2021) attributed the lack of cooling to the growing Antarctic ice sheet, which could have led to increased

stratification and shielding of deeper waters in the Southern Ocean. We here conclude that the southern South Atlantic regions already reached cold conditions during the MMCT and, as a result, could not cool much further given the global climate of the Late Miocene. The cooling in the subtropics (Sites 1088 and 1168) is in the Late Miocene much more pronounced than the southern South Atlantic, because of the gradual northward expansion of the westerly winds, ACC, and cold subantarctic waters (Leutert et al., 2020) in response to the expansion of the Antarctic ice sheet in the Late Miocene. This process likely also

further promoted cooling in other sectors of the Antarctic-proximal Southern Ocean.

## 5.3    South Atlantic SST gradient evolution

Combining the SST records for two time slices discussed above (Sect. 5.2 and 5.3) yields a unique insight into the long-term temperature trends in the South Atlantic Ocean. The SST records from the South Atlantic region show unidirectional temperature drops (~8°C) across the EOT, with a small degree of polar amplification between Antarctic-proximal (Site 696

and 511) and subtropical records (Site 1090). The strong, large and persistent South Atlantic Subpolar Gyre across the EOT kept the latitudinal SST gradient low in the southernmost part of the South Atlantic (Huber et al., 2004; Houben et al., 2019). The temperature gradient across the South Atlantic increases during the MMCT, due to high latitude cooling towards modern values, followed by a subdued cooling the Late Miocene. Meanwhile the subtropical front SST records (Sites 1090 and 1088) remain relatively stable from the earliest Oligocene (33 Ma) until the Late Miocene (Fig. 8), with minimal cooling until the

latest Miocene.



**Figure 8.** South Atlantic SST compilation A. Paleolatitude evolution of sites presented in this study (see legend; http://www.paleolatitude.org, version 2 by van Hinsbergen et al. (2015)). Stippled lines indicate hiatuses. B. The colored points and lines indicate the biomarker-based SSTs (see legend), excluding all samples with potential GDGT overprints (see Sect. 3.2). TEX$_{86}$-SST data from Site 696 (yellow; this study). TEX$_{86}$-SST data from Site 1090 (dark pink; Liu et al., 2009). U$^{k'}_{37}$ - and TEX$_{86}$-SST from Site 511 (green; Houben et al., 2019). U$^{k'}_{37}$ -SST from 1088 (pink; Herbert et al., 2016). TEX$_{86}$-SST from Site U1536 (red; this study). Clumped isotope ($\Delta_{47}$) SST estimates from La Meseta Fm., Seymour Island (purple; Douglas et al., 2014). C. Benthic foraminiferal $\delta^{18}$O compilation, with a locally weighted smooth over 20 kyr (black curve) (CENOGRID; Westerhold et al., 2020), and a LOESS smooth (blue; span = 0.2). D. Published paleo-CO$_2$ data (https://www.paleo-co2.org), with LOESS smooth (red; span = 0.2). LMC=Late Miocene Cooling, MCO=Miocene climatic optimum, EOIS=Early Oligocene oxygen isotope step, EOT=Eocene-Oligocene transition. The gray shaded areas indicate the time intervals we discussed in Sect. 5.1 and 5.2. These Sections are indicated on top of the figure.



The cooling phases in the South Atlantic across the EOT and from the MCO to the MMCT represent two climatic transitional
phases, both characterized by declining atmospheric $pCO_2$ concentrations (Pearson et al., 2009; Foster et al., 2012; Greenop
et al., 2014; Steinthorsdottir et al., 2016) and increasing benthic foraminiferal $\delta^{18}O$ values (Westerhold et al., 2020), indicating
deep-sea cooling and/or ice sheet expansion (Flower and Kennett, 1993; Zachos et al., 1996) (Fig. 8). The EOT marks the first
installation of a continent-wide Antarctic ice sheet (Deconto and Pollard, 2003; Coxall et al., 2005), with a volume between
60 and 130% of that of the present day ice sheet (Bohaty et al., 2012). The MCO is considered a global warm phase, with
warm-temperate ice proximal conditions (Sangiorgi et al., 2018) and a profoundly reduced Antarctic ice volume (Shevenell et
al., 2008; Foster et al., 2012), and the MMCT a strong, stepwise transition towards a larger Antarctic ice sheet (Rohling et al.,
2022). Surprisingly, southern South Atlantic records (Sites 696 and U1536) suggest similar Antarctic-proximal SSTs (~12–
14°C) for the early Oligocene, when a large, predominately terrestrial ice sheet, with marine terminating glaciers was installed,
as for the MCO, when ice sheets were profoundly reduced. Keeping in mind the higher-than-modern Antarctic
paleotopography in the Oligocene, with a gradual subsidence during the Miocene (Paxman et al., 2019), this still puts both
climate phases into perspective: apparently the Oligocene Antarctic ice sheet could coexist with warm ice-proximal surface
ocean conditions, while the Middle Miocene Antarctic ice sheet could be strongly reduced despite a relatively cold ice-
proximal South Atlantic Ocean.

## 6    Conclusions

Our lipid biomarker records from IODP Site U1536 and ODP Site 696 have generated new insights for the understanding of
the SST evolution of the South Atlantic Ocean, viz:

- The EOT in the South Atlantic is characterized by a relatively small latitudinal SST gradient of ~5 degrees between
  the subtropical front and the western Weddell Sea and a regional SST decrease (4–6°C) as global $pCO_2$ declined.
- South Atlantic SSTs show no prominent increase in the SST gradient across the EOT, which we ascribe to the
persistent gyral circulation that connects all South Atlantic sites, in the absence of a strong throughflow through Drake
  Passage.
- Southern South Atlantic SSTs at the earliest Oligocene glaciation were similar to those of the warm MCO, implying
  Antarctic proximal SSTs are not the only determining factor for the extent of the Antarctic ice sheet.
- The southern South Atlantic experienced cold polar climate conditions (SSTs of ~5°C) already during the MMCT.
This made it the coldest oceanic region around Antarctica and the likely region of deep-water formation.
- Due to the already relatively cold conditions in the southern South Atlantic in the Middle Miocene, it experienced
  little further cooling during the Late Miocene. This is in contrast to subtropical sites and other sectors of the Southern
  Ocean, which experienced profound cooling due to northward expansion of the Southern Ocean frontal systems as
  the Antarctic ice sheet expanded in the Late Miocene.




**Author contributions**

FSH, PKB and FS designed the research. FSH, PKB, CE, ALQ and JE collected the samples. ALQ ansd CE provided depositional information, core description, facies analyses and age constraint for ODP Site 696. SvdL advised on the Drake Passage tectonic evolution and provided (paleo) geographic maps for figures 1, 6 and 7. JE, MAS and FSH processed samples
for organic geochemistry. FSH, PKB, FP and FS interpreted the data. FSH wrote the paper with input from all authors.

**Competing interest**

The authors declare that they have no conflict of interest.

**Acknowledgements**

This work used International Ocean Discovery Program (IODP) archived samples and data. We thank the the great scientist and crew on Expedition 382, also helping with data interpretation and discussions of results from Site U1536. We thank Mariska Hoorweg for technical support at the Utrecht University GeoLab. FSH and PKB acknowledges funding from the NWO polar programme (grant number ALW.2016.001) and ERC starting grant 802835 "OceanNice". CE and ALQ acknowledges funding provided by the Spanish Ministry of Science and Innovation (grants CTM2014-60451-C2-1/2-P and CTM2017-89711-C2-
1/2-P, co-funded by the European Union through FEDER funds) and FJC2021-047046-I (MCIN/AEI/10.13039/501100011033 and NextGenerationEU/PRTR). SvdL acknowledges funding by NWO Vici (grant nr. 865.17.001) awarded to Douwe van Hinsbergen.

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
