# Peer review of "Late Cenozoic Sea Surface Temperature evolution of the South Atlantic Ocean"

_EGUsphere, 2023_

## Author Response (AR1)

**Response to reviewers: Late Cenozoic Sea Surface Temperature evolution of the South Atlantic Ocean, climate of the past**

**Reviewer #1**
**We thank the reviewer for their positive assessment, and their constructive feedback which we reply to (in bold) below each comment. Their input has improved the manuscript, which we have adapted considering most of their comments. We (will) have another thorough reading of the text to improve the language and avoid confusion.**

Hoem et al. have built the TEX86-SST records of the South Atlantic Ocean during two periods, from the late Eocene to early Oligocene (Site 696) and from middle to late Miocene (Site U1536). They compared the new SST records with the previously published SST records and linked the SST evolution of the Southern Atlantic Ocean to changes in ocean circulation. Overall, the manuscript is written well, but several phrases/sentences are confusing and difficult to understand.

The authors showed that the SST record of Site 696 across the EOT is similar to that from Site 511, and they ascribed the regional similarity in SSTs to a persistent, strong Subpolar Gyre circulation, connecting all sites, which can only exist in absence of a strong throughflow across the Drake Passage. This explanation sounds reasonable, but the authors did not explain clearly in the paragraph of discussing what caused the similar SST between Site 696 and 511. Further explanation is needed. Please look at the specific comments below.
**We have now clarified the concept with extra explanations at the end of paragraph 5.1 "discussion". The text on the similar SSTs across the South Atlantic now reads as follows:**

**"The persistently similar southern South Atlantic SSTs (~2°C difference) across the EOT would suggest that Sites 696 and 511 remained under the same surface ocean circulation regime. The similar SSTs implies throughflow across the opening Drake Passage did not change the ocean circulation across the EOT. A larger throughflow through Drake Passage would increase the temperature gradient between sites 696 and 511. Tectonic evidence suggests that the Drake Passage was narrow, with little deep-water connection from the Pacific to the Atlantic around the EOT (Livermore et al., 2007; Eagles and Jokat, 2014; van de Lagemaat et al., 2021). Recent model experiments (Huber et al., 2004; Hill et al., 2013; England et al., 2017; Sauermilch et al., 2021) show that a Southern Ocean without open gateways featured wind-driven clockwise gyres in the South Pacific and South Indian/Atlantic Ocean basins (Fig. 6B) that would advect warm surface waters toward the Antarctic coast. Specifically, eddy resolving ocean model simulations by Sauermilch et al., (2021) for the Eocene show that a restricted Drake Passage throughflow would sustain the persistent subpolar gyre and lead to SSTs reaching 19°C in the Australian-Antarctic Basin and 15–17°C in the subpolar Pacific and Atlantic. This is very similar to our TEX86-SST record at Site 696. We thus propose that the small SST difference between Sites 511 and 696 is the result of a restricted Drake Passage during the latest Eocene – earliest Oligocene, facilitating a persistent wind driven gyral circulation that connected the discussed southern South Atlantic sites.**

**In addition position of the continents, has the ice sheet extent and the position of the winds a strong effect on SSTs and patterns of Southern Ocean circulation (e.g., Scher et al., 2015; Evangelinos et al., 2022). Since the Antarctic proximal Site 696 shows relatively warm temperatures and a lack of a strong cooling it is very unlikely that the ice sheet extended to the marine realm where it would influence the SSTs. The southward shifts of westerly wind stress associated with the onset of Antarctic glaciation across the EOT (Kennedy et al., 2015) could expand the size of subtropical gyres and shrink the size of subpolar gyres (Xing et al., 2022). However, an atmospheric forced invigoration of the subpolar gyre (Houben et al., 2019) did not induce stronger latitudinal temperature gradients between these sites according to our data."**

I disagree with the authors' arguments about Miocene Climate Optimum (MCO) or any comparison with MCO, given that only three SST data points (one data point from Site U1536, two data points from Site 696) are available for the MCO (Figure 7A or Figure 8). SSTs within the MCO interval are likely to vary (for example, MCO SST from Sites 1168 and 1171 shown in Figure 7), and thus any conclusion based on only three data points does not make sense to me.

**We agree with the reviewer and decide to explicitly mention the low resolution and age uncertainty in the text to make the reader aware of potential biases and turned down the tone of the conclusions. However, we still see value in publishing and discussing these results despite their low resolution and age uncertainties. The ages from Site 696 and the general late Miocene of Site U1536 are robust enough to discuss the cooling step. We acknowledge the uncertainties and the low resolution and make only suggestions based on our data rather than draw solid conclusions.**

Specific comments

Line 19:  delete "e.g.,"
**We will do so accordingly.**

Line 39: change "meridional gradients" to "meridional SST gradients"?
**We have made this change.**

Line 54: "has been improved" is unclear. Please specify the improvement.
**We have changed the sentence to: "Furthermore, high resolution modelling exercises (England et al., 2017; O'Brien et al., 2020; Sauermilch et al., 2021; Nooteboom et al., 2022) show a large effect of the opening and depth of the Southern Ocean gateways on the Australian-Antarctic Gulf oceanographic conditions"**

Line 123: deeper than what?
**We meant to say that Site 696 was deepening along with the formation of the South Orkney Microcontinent (SOM) during the EOT. We have changed this line to: "Across the EOT, sediments at Site 696 reflect an increasingly distal and deeper environment as the SOM continued to deepen (López-Quirós et al., 2021)."**

Line 149: delete "it means that".

**We have made this change.**

Line 151: change "consists" to "consist".
**We have made this change.**

Line 160-170: It seems that the TLE from Sites 696 (polar and apolar fractions) and U1536 (apolar, ketone and polar fractions) were separated fractions using different methods/solvent. Is it possible that different separation methods result in different TEX86 value for the same sample? How does the different separation method affect the comparison of TEX86-SST between Site 696 and U1536?
**The Site 696 TLEs were generated at LOCEAN and those for Site U1536 at Utrecht University. An interlaboratory comparison that assessed the repeatability and reproducibility of the TEX86 revealed that differences in sediment extraction and workup procedures do not affect isoprenoidal GDGT relative abundances. As the TEX86 is based on specific GDGT ratio and not on a single compound concentration, the estimated SSTs is therefore not biased by the extraction/separation method performed in the two labs. Instead, variations in reported TEX86 values appeared to be mainly introduced by the type of mass spectrometer used for GDGT analysis (Schouten et al., 2013). Since the polar fractions from both sites were analyzed using the same HPLC-MS instrument at Utrecht University, the uncertainty on our TEX86-based SSTs can be considered negligible.**

Line 209-210: Did the authors apply these prior means (20°C for the Late Eocene–Early Oligocene interval and 15°C for the Miocene) to all the sites mentioned in this study or just the new SST records from Sites 696 and U1536?
**Yes, on all sites. We clarify this in the new text:**
**"For all new and existing TEX86 records discussed in this study, we applied a standard deviation of ±20°C and a prior mean of 20°C for the Late Eocene–Early Oligocene interval and 15°C for the Miocene".**

Line 224: "high sample-to-sample variability" is confusing. Please rephrase it.
**We have taken this part out, as we discuss the high amplitude in the SSTs two sentences further down.**

Line 239: "middle and top intervals of the record" is redundant. Changed to "intervals"?
**We made the recommended change.**

Line 280 and 281: "amplified cooling" is confusing. Please rephrase it.
Line 281-283: I don't think this is a solid argument. First, TEX86-SST was not always higher than UK-SST from the late Eocene to early Oligocene at Site 511. For instance, TEX86-SST was slightly lower than UK-SST across the EOT. Second, the TEX86-SST value is partly controlled by the value of the prior mean when applying BAYSPAR to convert TEX86 to SST. When a lower prior mean is used, its TEX86-SST may be lower.
**This section on Site 511 proxy comparison has now been taken out as suggested below.**

Line 274-283: I don't think this paragraph is needed. Other people not the authors of this manuscript built the TEX86-SST and UK-SST record at Site 511, and thus it's not worthy using

one paragraph to discuss the difference between TEX86-SST and UK-SST. The main point is to compare previous SST records with the new records built by this manuscript.
**We have removed this section.**

Line 285-286: Reference "Douglas et al., 2014" is not in the list of references.
**Reference added.**

Line 285-286: Douglas et al., 2014 only shows Δ47-SST and TEX86-SST not UK-SST. Please double check it.
**We have adjusted the sentence.**

Line 285-286: why do you use Δ47-SST alone not including TEX86-SST, since you mentioned similar TEX86-SST at the same site. Please consider including TEX86-SST at Figure 6.
**We did not include the data set since the Douglas et al., 2014 TEX86 dataset only has one data point younger than 36 Ma, which is when our records start. We wanted to include the Δ47-SST since this method is believed to reflect absolute SST, and the SST estimate is very similar to the TEX86-based SST derived from nearby site 696.**

Line 286-288: Comparing different SST proxies from the Seymour Island is not essential for this manuscript, same problem as the discussion about SSTs of Site 511.
**We have rephrased the sentence to include the regional reliability of the SST proxy to our data from nearby site 696.**

Line 291: please specify "similar paleoceanography regime".
**We have removed this sentence. We elaborate on the oceanographic regime in the paragraph below.**

Line 297: This paragraph is not connected to the previous one. Need a transition sentence.
**We shortened and moved this paragraph around. See new text above in reply to second comment.**

Line 297: "from about 35.7 Ma" to what age? Or you mean "at about 35.7 Ma"?
**We have taken out the discussion of the glauconite horizon.**

Line 299: what does "This process" refer to?
**We changed "this process" to "The invigorated bottom currents"**

Line 304-307: where is Powell Basin? Please show the location of Powell Basin on the map of Figure 6, otherwise the readers cannot understand your argument.
**We have taken out the discussion regarding the Powell basin.**

Line 302: It seems that the discussion before "Site 696 (~588.8 to 577.9 mbsf) also" does not include Site 696, but the authors did not mention what sites they are discussing at the beginning of this paragraph. It's very confusing.
Line 305: add references to "Stronger bottom currents were thought to be the result of the opening of the Powell Basin and the establishment of oceanic upwelling, which drove regional cooling and increased primary productivity".

Line 305-306: why would we expect that the tectonic opening of Powell Basin resulted in an increase in the SST gradient between Sites 696 and 511.

Line 306-307: Please explain why the opening of the Powell Basin can affect the two sites similarly, given that the two sites at different latitudes are far wary from each other.

**We have removed this section (line 305-307) about the Powell Basin from the discussion.**

Line 307: Please consider rephrasing "The constant SST gradient" since the SST gradient was not calculated and shown in any figures.

**We have rephrased the sentence to " The persistently similar southern South Atlantic SSTs (~2°C difference) across the EOT would suggest that Sites 696 and 511 remained under the same surface ocean circulation regime". It is now directly following the discussion on the temperature difference in the region to make it more coherent.**

Line 308: please specify "same oceanographic regime".

**We have added more context into this paragraph that it is referring to the sites being under the same gyral circulation pattern. See reply to the second comment.**

Line 309-310: Confusing. How does the change is the throughflow across the Drake Passage affect the SST difference between Site 511 and 696? How did the authors deduce that "the Antarctic ice sheets only developed on land and therefore did not exert an influence on SSTs".

**A larger throughflow through Drake Passage would increase the temperature gradient between sites 696 and 511. Since the Antarctic proximal Site 696 still shows relatively warm temperatures and a lack of a strong cooling it is very unlikely that the ice sheet extended to the marine realm.**

Line 315: what do "these models" refer to? I thought they represent models from Hill et al. (2013) and England et al. (2017) you mentioned before, but the references you cited here are Huber et al. (2004) and Sauermilch et al. (2021). Very confusing.

**We´ve put the references together to avoid confusion.**

Line 319: "This is similar to our TEX86-SST record" is problematic. 15–17°C in the subpolar Pacific and Atlantic from Sauermilch et al. (2021) is for the time interval before the EOT rather than the EOT. However, the authors are discussing the SST across the EOT throughout this paragraph.

**Yes, the model is based on Eocene conditions, but for the model, the configuration and depth of the gateways are a more important factors than the time interval. Setting the time interval to the Oligocene would not change the output of SST estimated and ocean circulation patters much. However, deepening of the gateways would.**

Line 315-321: The authors cited several analyses from model simulations, but the argument/point is not clear. How is the model simulation related to the results from this study? Sauermilch et al. (2021) was cited here, and this paper argued that the opening of gateways caused the cooling of the Antarctica surface waters from the late Eocene to early Oligocene. Obviously, this paper does not support the authors' argument "a restricted Drake Passage". However, the authors used this paper to support "the small SST difference

between Sites 511 and 696 is the result of a restricted Drake Passage during the EOT –
earliest Oligocene".
**We have rewritten this part of the discussion (last two paragraphs of section 5.1), see reply to second comment.**

**In this section we discuss how our temperate conditions (18-12°C) and similar SSTs between Site 511 and 696 (~2°C) could be caused by a persistent gyre due to a restricted Drake Passage. This is also what the model simulations of Sauermilch et al. show, i.e., that the gyre circulation will continue unless the gateway is deep enough for deep water throughflow.**
**We try to make this more clear in the text about the data model comparison.**
**New text: "Specifically, eddy resolving ocean model simulations by Sauermilch et al., (2021) for the Eocene show that a restricted Drake Passage throughflow would sustain the persistent subpolar gyre and lead to SSTs reaching 19°C in the Australian-Antarctic Basin and 15–17°C in the subpolar Pacific and Atlantic. This is very similar to our $TEX_{86}$-SST record at Site 696. We thus propose that the small SST difference between Sites 511 and 696 is the result of a restricted Drake Passage during the latest Eocene – earliest Oligocene, facilitating a persistent wind driven gyral circulation that connected the discussed southern South Atlantic sites."**

Line 330: what do "lack of a strong SST gradient" and "absence of a prominent increase thereof" mean? The previous sentences of this paragraph did not mention "lack of a strong SST gradient" and "absence of a prominent increase thereof".
**This refers to the discussion in the previous paragraph referring to the constant temperature difference between site 696 and 511 across the EOT. This paragraph has been cleared up. See text in second comment.**

Line 331: what do "all sites" refer to? Which sites?
**We have now specified this in the text.**

Line 343-344: I would add "Figure 1" along with all the sites here to remind the readers of the location of these sites.
**We refer now to Figure 1 in this sentence.**

Line 359-361: The authors are comparing the SST between Miocene climate optimum and middle Miocene climate transition, but the SST within these two periods is not from the same site. MCO SST is from Site 696 while MMCT SST is from Site U1536. Their comparison does not make sense. Although the authors claimed that "Both sites were located at comparable paleolatitudes during the Miocene (Fig. 7B)", they cannot treat these two sites as the same site. Figure 7B shows from Site 696 and U1536 have different paleolatitude and paleolongitude at 16 Ma. Therefore, their following arguments, which is based on the treatment of Sites 696 and U1536 as the same site, are not reasonable, unless the author provide robust evidence to show that Sites 696 and U1536 are at the same location during their study time interval.
**We believe that we do not treat these two sites as one, but as two separate sites that are in close vicinity. To clarify this, we have now added that there is a ~2 degree latitudinal difference in location between the sites to the text: "Both sites were located at**

**comparable paleolatitudes (albeit with a 2.5° latitudinal difference, Fig 8A) during the Miocene (Fig. 7B)."**

Line 373-375: The sentence is too long. Please consider reorganizing it.
**We split this sentence in two.**

Line 386: The first sentence is not useful for the following discussion in this paragraph. Please consider deleting it.
**We deleted the sentence.**

Line 390: Please specify "the subtropical front sites".
**We have specified the site names.**

Line 398: This sentence does not make sense. The BWT record of Site 747 does not have data for the late Miocene. Also, Site U1536 does not have BWT record. You cannot use "both".
**We have restructured the sentence to clarify the separate records;**
**"Sites U1536 SSTs and 747 BWTs reached cold temperatures by the latest-MMCT (Leutert et al., 2021), with minor cooling thereafter. Leutert et al. (2021) attributed the lack of post MMCT cooling to the growing Antarctic ice sheet, which could have led to increased stratification and shielding of deeper waters in the Southern Ocean"**

Line 409: what does "a small degree of polar amplification" mean?
**We have elaborated on this in the text; "The SST records from the South Atlantic region show unidirectional temperature drops across the EOT, with a small degree of polar amplification where Antarctic-proximal (Site 696 and 511) cooled by ~8°C and subtropical records (Site 1090) cooled by ~5°C."**

Line 412: which figure shows "The temperature gradient across the South Atlantic increases during the MMCT"?
**We now refer to both figure 7 and 8.**

Line 414: This argument is weak since Site 1090 does not have the SST record of the late Miocene.
**With sites 1088 and 1090 being relatively close to each other (<4 degrees latitudinal difference) and both in the vicinity of the subtropical front, we name these the "subtropical front sites", where Site 1090 records the SST of the STF during the Eocene-Oligocene and Site 1088 the SSTs during the late Miocene. We added the site number to the time intervals in the text to specify.**

Line 434: "considered as a warm phase" or "considered to be a warm phase"
**We changed to "as a".**

Line 436: change "the MMCT a strong, stepwise transition" to "the MMCT is a strong and stepwise transition"?
**We changed the text accordingly.**

Line 437-439: The claim that the Antarctic-proximal SST (Sites 696 and U1536) during the early Oligocene is same as that during the MCO is not convincing. Only three SST data points (one data point from Site U1536, two data points from Site 696) are available for the MCO (Figure 7A or Figure 8). Although the data points of the early Oligocene SSTs from Site 696 are adequate, few (ONLY TWO for Site 696) SST data for the MCO does not allow us to compare the SST between these two periods.

**Indeed, unfortunately we don't have many data points for the Miocene. However, we still see value in showing and discussing the data that we have as there is barely any existing data from this time for the South Atlantic. The age model and recent investigation by López-Quirós et al., in prep. does confirm a robust age range of 16.7-16.5 Ma (within MCO) for core 50R where the two datapoints are generated from.**

**We will include a sentence about the low resolution of the records and the uncertainties that this brings: "Surprisingly, although southern South Atlantic records (Sites 696 and U1536) are of low resolution with notable age uncertainties, they do suggest similar Antarctic-proximal SSTs (~12–14°C) for the early Oligocene, when a large, predominately terrestrial ice sheet with marine terminating glaciers was installed, as for the MCO, when ice sheets were profoundly reduced."**

Figure 6: the purple star on Seymour Island is hardly visible.
**We have increased the size of the star and changed the border to white to make it more visible.**

Figure 8: it is difficult to distinguish different CO2 proxies based on the current colors. Please change the colors or/and use different symbols.
**We have changed the color scheme and also removed some of the less reliable CO2 proxies from this figure.**

**Reviewer #2**

**We thank the reviewer #2 for their positive and constructive assessment of the manuscript and reply to each comment below. Their input has improved the manuscript, which we have adapted considering most of their comments.**
**See our response in bold text under each comment.**

In this paper, Hoem et al., present new lipid biomarker sea surface temperature reconstructions from the South Atlantic during the latest Eocene/earliest Oligocene and mid-to-late Miocene. TEX86 SST estimates from ODP Site 696 (Eocene/Oligocene-aged) show cooling from the late Eocene to early Oligocene, consistent with previous studies. However, they find high variability around the Eocene-Oligocene transition (SSTs ranging from ~5 to 22 degrees). TEX values from IODP 1536 (Miocene-aged) are affected by high terrestrial input – as such, this record is low-resolution. However, it does provide a snapshot into middle and late Miocene SSTs. Intriguingly, there is little cooling at this site between the mid and late Miocene. This differs to the SW Pacific, where there is significant cooling from the mid to late Miocene

This is an interesting paper and provides new insights into how Southern Ocean operated during the Eocene/Oligocene and Miocene. However, I have a few comments and suggestions below. I would also point the authors towards a couple of new papers that may be useful, namely Tibbett et al,. 2021 (P&P) and Tibbett et al., 2023 (P&P) for the Eocene/Oligoecne and Duncan et al., 2023 (Nat. Geo) for the Oligo/Miocene.
**Thank you for pointing this out. We have added references to Duncan and Tibbett papers.**

High variability in SSTs
I really like the high-resolution data across the EOT - 15 kyr resolution is great! However, you do see a lot of variability. Could this be a genuine climate signal? Could this perhaps capture orbital variability?
**We unfortunately do not have a strong enough age model to infer orbital cycles and discuss a genuine climate signal, which is why we do not go into a more detailed discussion on this part of the record.**

In the manuscript, you invoke upwelling as one potential hypothesis – is there any supporting evidence for changes in upwelling? Any changes in dinocyst assemblages?
**The Eocene- Miocene paleoecology using dinocysts in the southern South Atlantic is the subject of a paper that we are currently working on. Hopefully in preprint by the time this manuscript is released. We have added a sentence on this now; "Southern South Atlantic low-resolution Late Eocene-Early Oligocene palynological investigations (Houben et al., 2019; Hoem et al., in prep), shows a high diversity in the dinocyst assemblages with varying relative abundance, including Antarctic derived, open ocean, temperate and high nutrient indicative species respectively, inferring a fluctuation in surface ocean conditions potentially related to shifts in frontal systems and upwelling regions."**

Or is it due to other processes? In L260 you also mention big changes in ice rafted debris. Could some of this variability be explained by reworking and input of isoGDGTs from pre-

Eocene source rocks? Tibbett et al., 2021 also see big changes in reworking during the E/O at Prydz Bay (although the actual impact on TEX86 values seem to be minimal).

**We will include the Tibbett et al., 2021 discussion on reworking isoGDGTs on the 696 TEX86 record. The GDGT abundances in our Site 696 record (supplementary table and Figs 4b and S1) indicate low BIT index values (<0.2) throughout the record, suggesting little input from land. Hence, the influence of reworking on the GDGT pool in the sediment is likely lower at the location of Site 696 than in Prydz bay. We add the BIT index values of site 696 into the discussion.**

2. Atlantic SST gradients

You argue there is no change in SST gradients in the South Atlantic across the EOT. Figure 6 seems to support this - but then I re-read the manuscript, and its unclear if you are referring to changes in SST gradients between (a) the mid-latitude S. Atlantic site (ODP 1090) and the high-latitude S. Atlantic sites (696, 511) OR (b) between the entire South Atlantic (e.g,. 1090, 696, 511) and the equatorial Atlantic (e.g., 925). This needs to be clarified.

**The focus the late Eocene-early Oligocene discussion lies on the comparison of SSTs between the southern South Atlantic sites 511 and 696. We are now more careful with naming the specific sites and the regions we discuss in the in the last two paragraph of section 5.1. "The persistently similar southern South Atlantic SSTs (~2°C difference) across the EOT would suggest that Sites 696 and 511 remained under the same surface ocean circulation regime... "**

Moreover, it would also be very informative to actually SHOW the gradients and how they evolve through time. Are they stable or do you find subtle changes through time. This is important for supporting many of your arguments.

**Unfortunately the age models of the sites are of insufficient temporal resolution to do such a detailed point-by-point comparison. The focus of the manuscript is the general SST trends rather then the point to point comparisons in the respective datasets.**

3. Zonal mean SST gradients

During the Miocene, there is cooling in the South Pacific but not in the South Atlantic. You suggest this is because the South Pacific was already very cold, so couldn't cool any further. But why did the South Pacific cool? Was this due to declining CO2? Or some other factor? Perhaps some regional change (e.g, gateways) could explain why the South Atlantic remains stable and the South Pacific cools.

**Yes, we say that the South Atlantic already was cold in the latest-MMCT and did not cool further in the late Miocene, while the south Indian sites did cool further in the late Miocene (e.g., Herbert et al., 2016; Hou et al., 2023). This could be because the ice growth is not always uniform around Antarctica – the place with the largest cooling is the sector where the Antarctic ice sheet expands most, causing regional cooling and sea ice expansion. We suggest there was sea ice expansion and deep water formation in the southern South Atlantic (Site U1536) already by the end of the MMCT. However, since we lack high latitude records from across other regions of the Southern Ocean we will not go into further discussion at this point.**

4. Oligocene vs Miocene

At the very end, you note that SSTs appear similar in the Southern Atlantic in the Oligocene (lots of ice) and mid Miocene (less ice). However, there are only a few data points for the MCO so would be cautious in overspeculating.

**We have now emphasized the low resolution in this data comparison and the age model uncertainties in the manuscript, and explained the uncertainties that this brings for the interpretation of these records for this time interval.**

L110: as there is one age model tie-point in the Miocene for ODP Site 696, how confident are the estimates shown in Figure 7 panel a.

**With respect to Site 696, the study interval for this work includes sediments recovered from 607.6 - 520.2 mbsf (cores 62-50). Our age marker (tie point) for the lower Miocene interval comes from core 50 (at 521 mbsf). Although our paper refers to a single age marker (from core 50), the entire lower Miocene interval (cores 46-50; 478 - 520 mbsf) was originally constrained based on diatom biostratigraphy by the *N. grossepunctata Zone*: 14.3 - 14.8 Ma; Barker et al., 1988; Gersonde and Burckle, 1990). After drilling time, diatom biostratigraphic schemes were adjusted/updated (e.g., Carter et al. (2017) indicated a refined age of 17.6 - 15.4 Ma for core 50). On the basis of a more thorough review, the lower Miocene section (Core 50) should be dated to about 16.5 - 16.7 Ma (López-Quirós et al. in prep). Consequently, the estimates shown in Figure 7 panel a are robust and ascribed to a Mid Miocene time.**

**References:**
**Gersonde, R., and Burckle, L.H. (1990) Neogene Diatom biostratigraphy of ODP Leg 113, Weddell Sea Antarctic Ocean. In P.F. Barker and J.P. Kennett et al., Eds., Proceeding of the Ocean Drilling Program, Scientific Results, Leg 113, vol. 113, p. 761–789. Ocean Drilling Program.**

**Barker, P.F., and Kennett, J.P., et al. (1988) Proceedings of the Ocean Drilling Program, Initial Reports, 113. Ocean Drilling Program, College Station, Texas.**

**Carter, A., Riley, T.R., Hillenbrand, C.-D., Rittner, M., 2017. Widespread Antarctic glaciation during the late Eocene. Earth Planet. Sci. Lett. 458, 49–57.**
**Case, J.A., 1988. Paleogene floras from Seymour Island, Antarctic Peninsula.**

L163: make sure to define any abbreviations. This occurs throughout the analytical methods section.

**We have added the full words to the abbreviations.**

Figure 7: you show data from South Atlantic and SW Pacific, but the map only shows the South Atlantic. Please include the full region in the map.

**We now have a circum Antarctic map showing all the sites discussed.**

Figure 7: found it quite hard to discern the different colours especially 696 vs 1168 – similar colours!

**We have adjusted the color of Site 1168 to make it more visible.**

Figure 8: there is a lot of uncertainty surrounding CO2 proxies, esp. C3 plants, paleosols etc. Even alkenones are highly debated (see rae et al,. 2021). Would reconsider which CO2 proxies are most robust and show those. Or show all and mention caveats.
**We have removed the C3 plants and paleosols CO2 proxies from the plot.**

---

## Author Response (AR2)

Second round revisions for "Late Cenozoic Sea Surface Temperature evolution of the South Atlantic Ocean"

We thank the reviewer for their positive and constructive feedback. We respond to the individual comments in purple below and have made the appropriate edits to the manuscript.

The revised manuscript is well structed and clear. The language is fluent but not precise at some places. For instance, in the abstract "little additional cooling during the Late Miocene" is not consistent with "3–5°C SST decrease at Site U1536 between the MMCT (~14 Ma) and the latest Miocene (~6.4–5.2 Ma)" at Line 390-391. Please look at the specific comments below.
We have rephrased the line in the abstract to "relatively moderate cooling".
Cooling is small compared to mid latitudes which cooled by 10°C.

In addition, the authors did not explain why Site U1536 experienced reduced cooling during the late Miocene compared with the other sites from the Southern Ocean. The authors just concluded "Due to the already relatively cold conditions in the southwestern South Atlantic in the Middle Miocene, it experienced little further cooling during the Late Miocene". It seems that the authors assume that SST at this site cannot have additional cooling at a cold baseline. Why?
Since the lower end of the temperatures after the MMCT at Site U1536 were ~5-7°C, we argue that there is a limited amount these already cold temperature could cool compared to the warmer subtropics (around 24°C at MMCT). We have added some more information to make in more clear in line 405. There is no Site U1536 record between ~14.2-8.4 Ma (reflector-c) where we suggest a prolonged time interval of slow sedimentation rates or non-deposition or erosion cause the gap. Temperatures could´ve been colder and variable in this time, and sea ice might have been present, but without the record we cant speculate any further.

Specific comments:
Line 24-25: "the regional similarity in SSTs" is confusing since you did not mention it earlier. Add some info about similar SSTs between Site 696 and 511 before this sentence?
We have changed the sentence accordingly.

Line 65: delete "occurred that".
We have changed the sentence accordingly.

Line 68: "during this time" is confusing. Please specify "this time".
We changed "this time" to "since the late Eocene".

Line 221-222: change "refer to the proxy results as such in the reminder of this work" to "refer to the proxy results as SSTs in the reminder of this work".
We have changed the sentence accordingly.

Line 225-227: In your supplementary information, Fig. S3 shows that BAYSPAR-derived SSTs are higher than the SSTs derived from Kim et al. (2010) and O'Brien et al. (2017). Please check if the markers used for different calibrations in Fig. S3 are correct or not.
Yes the colors were changed between the two calibrations. This has now been corrected.

Line 224: Kim et al. (2010) generated two calibrations from GDGTs to SST. Does this study use GDGT index-1 or GDGT index-2?
We used index-2. I´ve included this in the text.

Line 225: I don't think the calibration from O'Brien et al. (2017) designed for warm Cretaceous climates is appropriate for this study. O'Brien et al. (2017) excludes all data for which satellite-derived mean annual SSTs are < 15 °C, probably including the regions where your study sites are located.
We argue to still keep this calibration in as a reference. Although the calibration is not specifically made for Oligocene-Miocene conditions, doesn't mean it is wrong at lower temperatures. It is very close in absolute temperature to the linear calibration from Kim et al., 2010.

Line 265-297: This paragraph is too long. Please consider breaking it into two paragraphs.
We split this paragraph into two.

Line 272: Please cite the most recent review paper for Cenozoic CO2, Rae et al. (2021)
https://doi.org/10.1146/annurev-earth-082420-063026
We´ve added this reference.

Line 272: Hoenisch, 2021 is not in your list of references. Please check if you put all the cited references in the reference list.
This ref has been added. We will go through and check.

Line 453: please specify "SST gradient". Latitudinal SST gradient or gradient between Site 511 and 696?
We also mean Site 1090. I now added "latitudinal" before SST gradient.

Line 457: delete "already".
We have deleted this word.